# Evolution of fungal and non-fungal eukaryotic communities in response to thermophilic co-composting of various nitrogen-rich green feedstocks

**Felix Matheri**[1,2]*, **Anne Kelly Kambura**[3], **Maina Mwangi**[1], **Edward Karanja**[2], **Noah Adamtey**[4], **Kennedy Wanjau**[5], **Edwin Mwangi**[2], **Chrysantus Mbi Tanga**[2], **David Bautze**[4], **Steven Runo**[1]

**1** Department of Biochemistry, Microbiology, and Biotechnology, Kenyatta University (KU), Nairobi, Kenya, **2** International Centre for Insect Physiology and Ecology (icipe), Nairobi, Kenya, **3** Department of Agricultural Sciences, Taita Taveta University (TTU), Voi, Kenya, **4** Research Institute of Organic Agriculture (FiBL), Frick, Switzerland, **5** International Livestock Research Institute (ILRI), Department Animal and Human Health, Nairobi, Kenya

* fmatheri@icipe.org

**Data Availability Statement:** The raw ITS and 18S sequences were submitted to the NCBI sequence

## Abstract

Thermophilic composting is a promising soil and waste management approach involving diverse micro and macro-organisms, including eukaryotes. Due to sub-optimal amounts of nutrients in manure, supplemental feedstock materials such as Lantana camara, and Tithonia diversifolia twigs are used in composting. These materials have, however, been reported to have antimicrobial activity in in-vitro experiments. Furthermore, the phytochemical analysis has shown differences in their complexities, thus possibly requiring various periods to break down. Therefore, it is necessary to understand these materials' influence on the biological and physical-chemical stability of compost. Most compost microbiome studies have been bacterial-centric, leaving out eukaryotes despite their critical role in the environment. Here, the influence of different green feedstock on the fungal and non-fungal eukaryotic community structure in a thermophilic compost environment was examined. Total community fungal and non-fungal eukaryotic DNA was recovered from triplicate compost samples of four experimental regimes. Sequencing for fungal ITS and non-fungal eukaryotes; 18S rDNA was done under the Illumina Miseq platform, and bioinformatics analysis was done using Divisive Amplicon Denoising Algorithm version 2 workflow in R version 4.1. Samples of mixed compost and composting day 84 recorded significantly (P<0.05) higher overall fungal populations, while Lantana-based compost and composting day 84 revealed the highest fungal community diversity. Non-fungal eukaryotic richness was significantly (P< 0.05) more abundant in Tithonia-based compost and composting day 21. The most diverse non-fungal eukaryotic biome was in the Tithonia-based compost and composting day 84. Sordariomycetes and Holozoa were the most contributors to the fungal and non-fungal community interactions in the compost environment, respectively. The findings of this study unravel the inherent influence of diverse composting materials and days on the eukaryotic community structure and compost's biological and chemical stability.

archive with accession number PRJNA822850
(https://www.ncbi.nlm.nih.gov/sra/PRJNA822850).

**Funding:** This research received partial financial
support from Biovision Foundation (Grant No:
1040), Coop Sustainability Fund (Grant No: 1040),
Liechtenstein Development Service (LED) (Grant
No: 1040) and Swiss Agency for Development and
Cooperation (SDC) (Grant No: 1040),. The authors
also acknowledge partial financial support from the
DAAD-Africa through the "In-Country/ In-Region
Scholarship Programmes Eastern Africa" (Grant
No: 91712524) We also gratefully acknowledge the
support of icipe core funding provided by United
Kingdom's Foreign, Commonwealth and
Development Office (FCDO); the Swedish
International Development Cooperation Agency
(Sida); the Swiss Agency for Development and
Cooperation (SDC); the Federal Democratic
Republic of Ethiopia; and the Government of the
Republic of Kenya.The funders had no role in study
design, data collection and analysis, decision to
publish, or preparation of the manuscript. The
funders had no role in study design, data collection
and analysis, decision to publish, or preparation of
the manuscript".

**Competing interests:** The authors have declared
that no competing interests exist.

# Introduction

Thermophilic composting, also known as hot rotting, is the degradation of organic material driven by different categories of organisms that bring about turns of high and low temperatures [1, 2]. This composting method is ideal for managing organic wastes from systems such as agricultural, municipal, and food industries that would otherwise be hazardous to the environment [3]. Microorganisms involved in composting include prokaryotes, protozoa, fungi, and other eukaryotes, with prokaryotes being dominant and most studied [4, 5]. Though less studied, eukaryotes are an important microbial category in the ecosystem with diverse feeding and genetic guilds [6, 7]. Eukaryotes degrade recalcitrant material such as Carbon-rich polymers, thus contributing to nutrient cycling in the compost environment [8]. The complex structure and function of a large number of compost microorganisms are directly affected by various environmental factors such as temperature, moisture, Carbon/Nitrogen ratio, and pH, among others [9]. Consequently, compost eukaryotes represent a largely unexplored frontier in microbial ecology and hold inordinate potential for discovering new communities and functional assemblages.

In agricultural systems, cattle manure is the primary composting material due to its ready availability and its harboring of important nutrients and microbes for plant growth [10]. However, due to suboptimal levels of important nutrients in cattle manure during composting, the common practice requires supplemental feedstock material. Green materials such as fresh grass clippings and fresh twigs of Lantana camara and Tithonia diversifolia have been recommended and used to supplement nitrogen levels in soils and compost [11, 12]. Some of these materials have been reported under invitro studies as having inhibitory properties on fungal and non-fungal organisms [13, 14]. The phytochemical complexities of Lantana camara and Tithonia diversifolia have been previously studied. Lantana camara has been reported to contain more complex polymers and thus possibly requires more microbial categories to break down into agriculturally useful material. For example, the phytochemical composition analysis of L. camara showed 23.3% crude fiber, 26% cellulose, 16.2% lignin, and 21% hemicellulose, while T. diversifolia contains 11.2%, 17%, 7%, and 16% of these elements respectively [15–17]. Furthermore, the different complexities of the composting material could require assorted composting times to produce a biogeochemically stable product. It is, therefore, necessary to evaluate the influence of composting time on the biological and physical-chemical quality of compost.

Numerous studies have assessed the prokaryotic community in the composting process using culture-dependent and culture-independent methods [2, 4, 6, 10]. However, there is still a limited understanding of eukaryotic communities' structure in the composting process, concerning the composition and complexity of the composting materials especially utilizing high-throughput sequencing technology. The effect of the Lantana camara and Tithonia diversifolia on microbial community structure in complex ecosystems such as the compost environment has not also been done, despite their widespread adoption by farmers as soil nutrient amendment materials.

This study comprehensively assessed fungal and non-fungal eukaryotic communities associated with the co-composting of assorted nitrogenous green material and composting time. The study hypothesized that different composting materials and days influenced eukaryotic communities differently. The culture-independent, high-throughput sequencing Illumina MiSeq platform was used for library preparation of the fungal and non-fungal eukaryotic communities in the different compost environments. Furthermore, the study evaluated the correlation and co-dependence of compost physical-chemical factors and their influence on fungal and non-fungal eukaryotic communities.

## Materials and methods

### Compost heaping and sampling

The composting materials that were common to all treatments were sourced from the same farm around the composting site. Particularly, raw manure for all the compost treatments was obtained from a single dairy unit. Compost preparation and heaping were done on the same day in the Long-Term Farming Systems Comparison trial site at Thika, Kenya (01 ˚ 0.231' S 37 ˚ 04.747' E) [www.system-comparison.fibl.org, 18]. The site was established by the Research Institute of Organic Agriculture (FiBL) and local partners, including the International Centre for Insect Physiology and Ecology (ICIPE) and Kenya Agricultural and Livestock Research Organization (KALRO).

Treatments were based on common farmer practices in Kenya and available sources of nitrogenous material for composting in the region [18, 19]. All compost treatments had an equal ratio of 4:2:1 w/w for fresh cow dung manure, dry maize stalks, and nitrogenous green materials, respectively. The different sources of nitrogenous green material treatments were as per Table 1.

Composting materials described above were individually cut into small pieces (3–5cm long) to enable uniform and faster breakdown. Compost heaping was done in triplicates for each of the four compost treatments. Each heap was done by laying small dry twigs on a flat leveled composting surface, followed by a layer of dry chopped maize stalks. Cow dung manure from zero-grazed cattle was used for the next layer, and, finally a layer of green material (Lantana/Tithonia/Grass/ mixture of Lantana, Tithonia, and Grass). The heaping process was repeated four times for each compost pile and heap moisture content was adjusted to about 60%, at the beginning of composting as recommended by [20]. Compost heaps aeration was done by turning every four days during the first 20 days and weekly for the following days till 84 days of composting. Sampling was done every 21 days, with the first 21 days of composting serving as the baseline. Therefore, sampling was done at 21, 42, 63, and 84 composting days.

### Physical-chemical characterization of compost treatments

The daily temperature was monitored using a compost thermometer (model: WIKA 110824862-EN 13190; Louisville, USA) at three locations on the heap by inserting the thermometer halfway between the top and bottom of the heap to the maximum probe depth (45 cm). Sampling for other physical-chemical parameters was done on the sampling days described above. The pH of the compost (1:10 w/v waste: water extract), moisture, germination index, and Carbon dioxide emission during sampling days (mg $CO_2$ g-1d-1) were done as described by [21]. Total Kjeldahl nitrogen (TKN) was analyzed using the Kjeldahl method. The total organic Carbon and total phosphorus (TP) (Olsen P) were analyzed according to [22].

**Table 1. Compost feedstock and green material types informing the treatments used for the experiment.**

| Materials used | Treatment/label |
| --- | --- |
| Fresh cow dung + dry maize stalks + fresh *Lantana camara* twigs | Lantana-based compost (L) |
| Fresh cow dung + dry maize stalks + fresh *Tithonia diversifolia* twigs | Tithonia-based compost (T) |
| Fresh cow dung + dry maize stalks + fresh grass clippings | Grass-based compost (G) |
| Fresh cow dung + dry maize stalks + fresh grass clippings, fresh *Tithonia diversifolia*, and *Lantana camara* twigs in the ratio of 1:1:1 | Lantana, Tithonia, Grass (Mixed)-based compost (LTG) |

*Wood ash (1kg) and soil (5kg) were sprinkled after every layer was heaped.

### Sampling for ITS and 18S rDNA analysis of compost eukaryotic communities

Compost samples for total DNA extraction were collected on days 21, 42, 63, and 84 of the composting process. Two set-sampling was done on each triplicate heap of each treatment from five different positions using a sharp shovel that was, pre-cleaned with 70% ethanol as described by [20, 23]. The samples were transported to ICIPE and stored at -20˚C. Total DNA extraction of the samples was carried out at the Kenyatta university plant transformation laboratories, Nairobi, Kenya.

### Compost microbiome total DNA extraction and amplification

Total compost DNA was extracted from triplicate compost subsamples of each treatment for fungal and other eukaryotic communities DNA extraction. The PureLink™ Microbiome DNA Purification Kit (Catalog number: A29790) as per the manufacturer's instructions (www. thermofisher.com/ke/en/home/life-science). Each extracted sample's quality and concentrations were measured under 2% agarose gel and NanoDrop (Maestrogen). The purified compost DNA was shipped under dry ice to the Molecular Research DNA Lab (www.mrdnalab. com, Shallowater, TX, USA) for sequencing under the illumine miseq platform.

The PCR primer sets used were ITS1-F (5'-CTTGGTCATTTAGAGGAAGTAA-3') ITS2 (5'-GCTGCGTTCTTCATCGATGC-3') for fungal ITS amplification while EUK1391F (5'GTACACACCGCCCGTC-3') and EukBr (5'-TGATCCTTCTGCAGGTTCACCTAC-3') for other eukaryotes. Amplification was done in 30 cycles PCR (5 cycles used on PCR using the HotStarTaq Plus Master Mix Kit (Qiagen, USA). The PCR conditions were 95˚C for 5 minutes, followed by 30cycles of 95˚C for 30 seconds, 53˚C for 40 seconds, and 72˚C for 1 minute, with a final elongation step at 72˚C for 10 minutes performed. Resultant PCR amplicons were visually quantified under 2% agarose gel. There were no fungal amplicons associated with grass-based compost sampled at 42 days. Equimolar quantities of PCR amplicons obtained from the remaining 31 samples (16 non-fungal eukaryotic amplicons from individual composts and 15 fungal amplicons) were multiplexed using unique indices, pooled, and sequenced using Illumina MiSeq next-generation technology at MR DNA (www.mrdnalab.com, Shallowater, TX, USA). Barcodes and amplicon primer sequences were trimmed after sequencing. Low-quality sequences were denoised and filtered out with reads <300 base pairs after phred20-based quality trimming. Sequences with ambiguous base calls and those with homopolymer runs exceeding 5bp were removed [24]. The raw ITS and 18S sequences were submitted to the NCBI sequence archive with accession number PRJNA822850 (https://www.ncbi. nlm.nih.gov/sra/PRJNA822850).

### Bioinformatics analysis

The Divisive Amplicon Denoising Algorithm 2 (DADA2) version 1.20.0 analyzed in R (version 4.1) as described by [25] was used as the primary workflow for analysis. Here, filtering of the reads, learning error rates, dereplication, merging of forward and reverse reads, chimera removal, and taxonomic assignment were done separately for the fungal (ITS) and eukaryotic (18S) sequences. The Unite reference database was used for ITS while, Silva 128 and 132 databases were used for non-fungal eukaryotes (18S) referencing. The products of the workflows were ITS and 18S Amplicon sequence variant (ASV) tables for the two amplicon sets [26].

### Statistical analysis

Data analysis was done using different packages in R version 4.1.2. The data for the various physical-chemical variables were individually subjected to a normality test using the Shapiro

test before means separation using ANOVA under the agricolae package (version 1.3–5). Comparing the compost treatments was done per composting day for each physical-chemical parameter, followed by a Tukey posthoc. The distributions of soil physicochemical variables across different compost treatments and composting days were calculated on log-standardized data using the "decostand" function in the Vegan version 2.6–2 package. The resulting distance matrix between samples was plotted in a PCA graph, with the projected direction and magnitude of the distribution for each variable plotted in a separate loading plot. This was followed by the computation of the Pearson correlation matrix using the function "corrplot" in the Corrplot package (version 0.92).

Alpha diversity metrics were calculated using the "estimate_richness" function in phyloseq Observed index was estimated to reflect the number of ASVs in each compost sample and values that have a positive correlation with the species richness. Shannon and Simpson Inverse-Simpson indices of the correlation of diverse species abundance in a sample [27] were also computed. Shapiro and significance tests were calculated before plotting the alpha diversity metrics. The taxonomy table and abundance table were merged with the abundance table, and bar plots of compost treatments and sampling day relative abundance were plotted using the "plot_bar" function. Venn diagrams to show the shared community ASVs among composting treatments and composting days were plotted using the eulerr (version 6.1.1) package. Beta diversity was computed using the Bray-Curtis index to further explore the influence of different composting treatments and composting days and differences on microbial community profiles. The resulting scores were used for PCoA plotting and compared using the PERMA-NOVA test of significance. A stepwise modeled Canonical Correspondence Analysis (CCA) was done in a Vegan package (version 2.6–2) to show the effect of explanatory physical-chemical variables on the different compost microbiomes A co-expression network detailing the interaction of abundant composting microbiomes was constructed based on microbial abundance using the "plot_net" function in the ggplot2 package (version 3.3.5).

## Results

### Different compost treatments exhibit distinct physical-chemical properties

Samples from the different compost treatments and composting days were shown to be significantly different (p-value ≤0.05) in terms of most physicochemical properties of treatments in most composting days. There were however notable similarities in the treatments, especially at the latter composting period (Table 2).

Physical-chemical nature of compost samples revealed significantly distinct compost properties for different composting materials and days (p-value < 0.05). The first principal component (PC1) contributed to 41.7% of the total variance while, the second principal component (PC2), contributed 22.1%. Together, the first two principal components accounted for 69.8% of the total physical-chemical variation of different compost treatments. Notably, day 21 of composting was highly distinct from other composting days. The most variability within groups was recorded in the Tithonia-based compost and composting day 42; Fig 1A and 1B.

On the other hand, most compost physical-chemical properties were correlated. The compost temperature was positively correlated with Carbon, Phosphorous, and Nitrogen but negatively correlated with pH and Nitrates (Fig 2).

### Different composting materials and days exhibit distinct biodiversity

Mixed compost (LTG) and composting day 84 recorded significantly (P<0.05) higher overall fungal populations ("Observed") among the compost types and composting days. Lantana-

**Table 2. Physical-chemical characteristics of different compost treatments on various composting days.**

| Component | Time | G | L | LTG | T | Trt |
|---|---|---|---|---|---|---|
| Organic Carbon (%) | 21 | 22.73[c] | 28.00[a] | 26.40[b] | 25.77[b] | *** |
| | 42 | 17.67[a] | 17.17[ab] | 16.07[bc] | 15.33[c] | ** |
| | 63 | 16.40[a] | 15.57[ab] | 15.20[ab] | 14.47[b] | * |
| | 84 | 13.75[a] | 13.67[a] | 12.87[c] | 13.27[b] | *** |
| Potassium (%) | 21 | 1.42[b] | 1.52[b] | 1.58[ab] | 1.78[a] | ** |
| | 42 | 1.26[a] | 1.16[b] | 1.19[ab] | 1.21[ab] | * |
| | 63 | 1.07[a] | 1.10[a] | 1.10[a] | 1.09[a] | ns |
| | 84 | 1.09[a] | 1.04[b] | 1.01[b] | 1.09[a] | *** |
| Total Nitrogen (%) | 21 | 0.77[b] | 0.96[a] | 0.82[ab] | 0.97[ab] | * |
| | 42 | 0.75[a] | 0.68[ab] | 0.67[ab] | 0.59[b] | * |
| | 63 | 0.62[a] | 0.58[a] | 0.59[a] | 0.49[a] | ns |
| | 84 | 0.55[a] | 0.60[a] | 0.54[a] | 0.55[a] | ns |
| Total Phosphorous (%) | 21 | 0.18[b] | 0.21[ab] | 0.24[a] | 0.22[ab] | * |
| | 42 | 0.18[a] | 0.14[b] | 0.13[b] | 0.14[b] | *** |
| | 63 | 0.16[a] | 0.15[a] | 0.15[a] | 0.15[a] | ns |
| | 84 | 0.17[a] | 0.16[b] | 0.14[c] | 0.16[ab] | *** |
| Moisture Content (%) | 21 | 72.2[a] | 73.9[a] | 70.9[a] | 72.3[a] | ns |
| | 42 | 46.7[b] | 51.1[a] | 44.9[c] | 51.6[a] | *** |
| | 63 | 17.1[a] | 18.8[a] | 19.6[a] | 18.8[a] | ns |
| | 84 | 18.3[a] | 19.3[a] | 19.3[a] | 19.8[a] | ns |
| Temperature (˚C) | 21 | 41.3[a] | 39.7[a] | 38.3[a] | 38.3[a] | ns |
| | 42 | 24.3[ab] | 23.0[b] | 26.3[a] | 25.0[ab] | ** |
| | 63 | 25.0[a] | 25.0[a] | 25.0[a] | 27.0[a] | ns |
| | 84 | 22.7[a] | 22.7[a] | 23.0[a] | 22.3[a] | ns |
| pH | 21 | 8.41[a] | 8.50[a] | 8.48[a] | 8.48[a] | ns |
| | 42 | 8.95[ab] | 9.00[a] | 8.90[b] | 8.99[a] | ** |
| | 63 | 8.37[a] | 8.36[a] | 8.25[a] | 8.47[a] | ns |
| | 84 | 8.72[a] | 8.62[c] | 8.55[d] | 8.67[b] | *** |

Values with the same superscripts were not significantly different, ns-not significant

* $P \leq 0.05$

** $P \leq 0.01$ and

*** $P \leq 0.001$.

based compost and composting day 84 had the highest fungal community diversity (Shannon, Simpson, and InvSimpson) (Fig 3A and 3B).

Non-fungal eukaryotic richness was significantly ($P < 0.05$) more abundant in tithonia-based compost (T) and composting day 21. The most diverse non-fungal eukaryotic biome was in the Tithonia-based compost and composting day 84 (Fig 4A and 4B).

## Different composting treatments and days influence the beta diversity of fungal communities distinctly

Overall, the most abundant fungal classes in the compost environment were Sordariomycetes (61% mean relative abundance), Agaricomycetes (8% mean relative abundance), Dothidiomycetes (20% mean relative abundance), Eurotiomycetes (4% mean relative abundance) and Saccharomycetes (2% mean relative abundance); Table 3. The fungal class Sordariomycetes was the most abundant taxa within the compost treatments, particularly in Tithonia-based

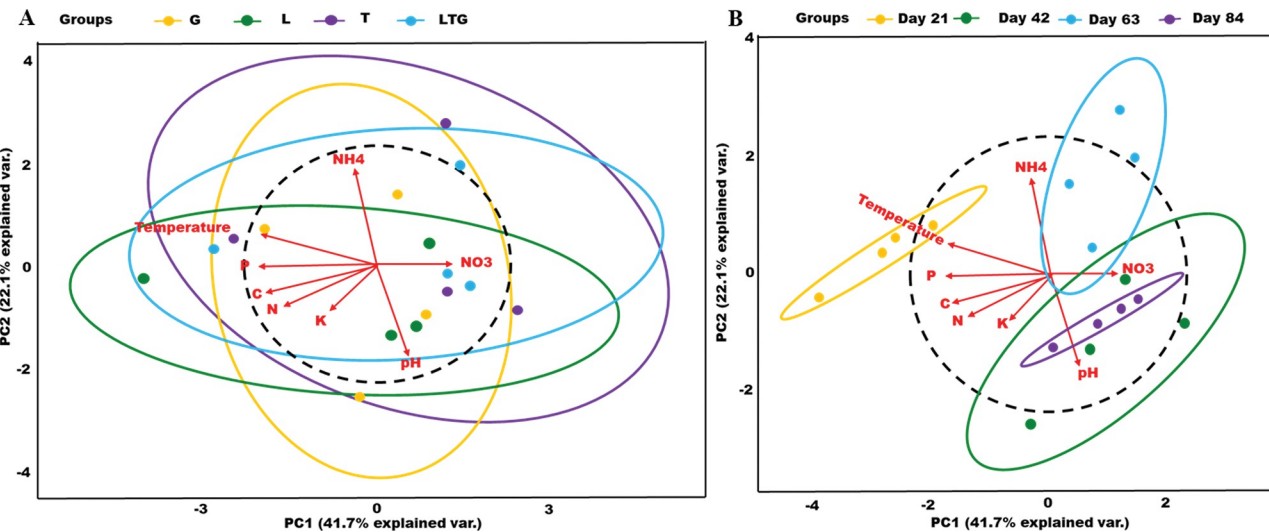

**Fig 1. A, B:** Principal component analysis (PCA) biplots of compost treatment (A) and composting days (B) according to their physical-chemical properties. Samples were clustered as per different compost types and composting days.

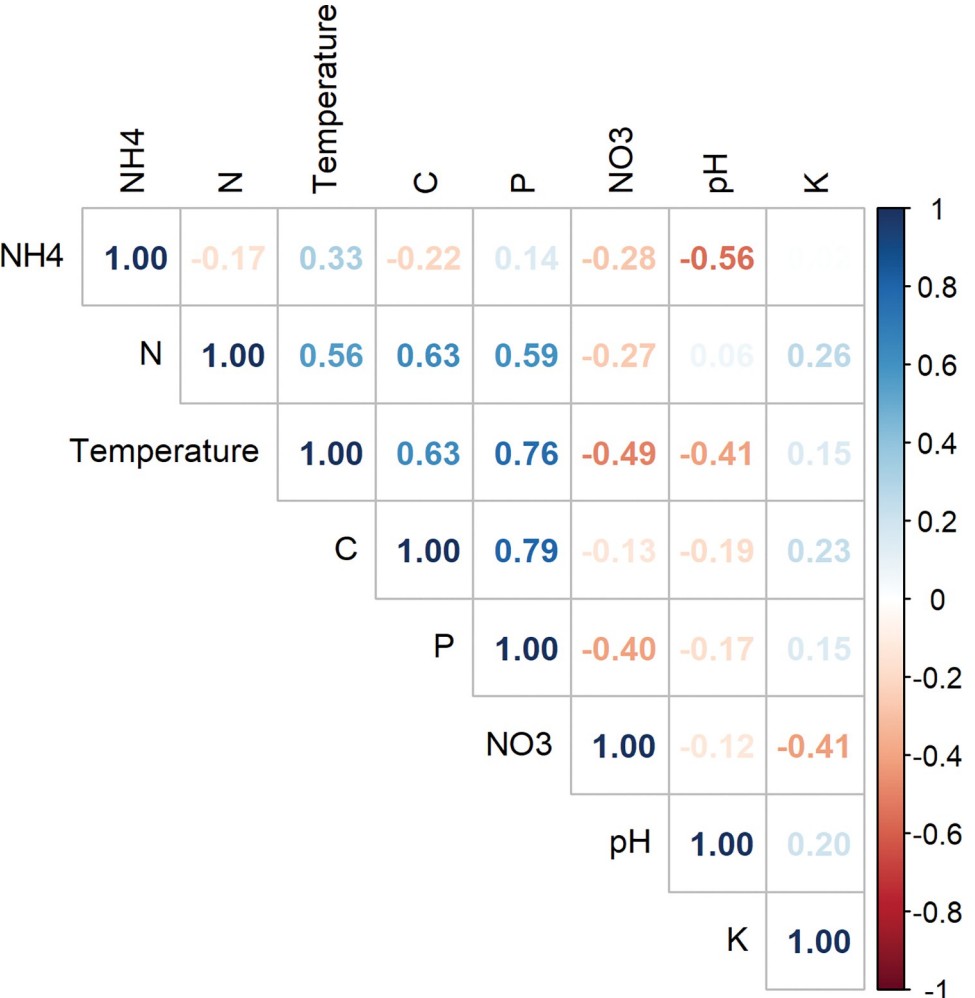

**Fig 2. Pearson correlation matrix between different physical-chemical variables (C); positive and negative correlations are displayed in blue and red, respectively.**

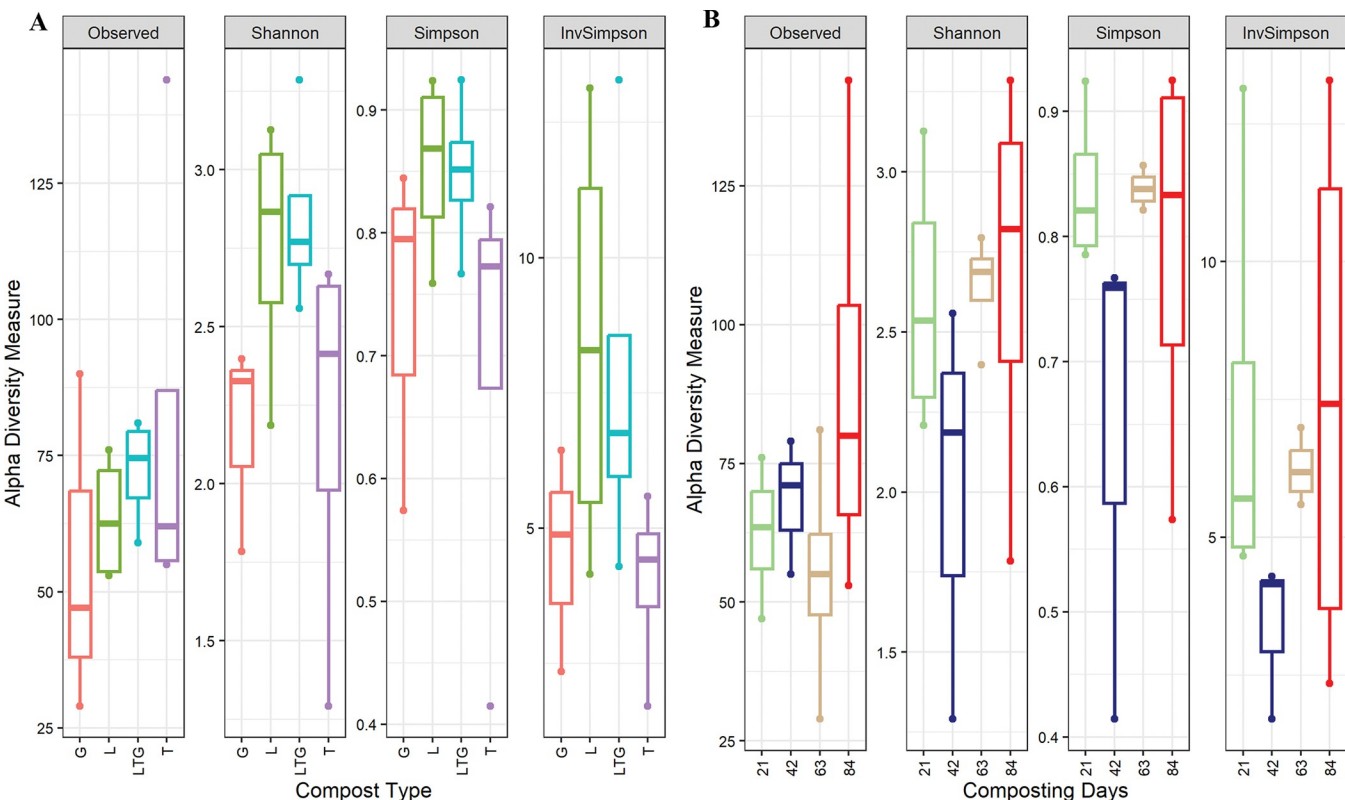

**Fig 3.** Alpha diversity metrics (Observed, Shannon, InvSimpson, and Simpson) of fungal eukaryotic communities under different composting treatments (A) and composting days. L, T, G, and LTG represent Lantana, Tithonia, Grass, and mixed (Lantana + Tithonia + Grass) based composts respectively. Day 21, day 42, day 63, and day 84 represent the effect of combined compost treatments at 21, 42, 63, and 84 days of composting.

composts which had a mean relative abundance of 68%; Fig 5A, and Table 3. Sordariomycetes was still the dominant fungal class on all the composting days with the highest abundance recorded at composting day 42 (74%). The class Agaricomycetes was most abundant at composting day 21 compared to other composting days, recording (Fig 5C and Table 3).

The most abundant non-fungal eukaryotic taxa in all the compost environments were Alveolata (4%), Chloroplastida (13%), Holozoa (58%), Rhizaria (13%), Stramenopiles (8%) and Tublinea (2%); Fig 5B, 5D and Table 4. Non-fungal eukaryotic class, Holozoa was the most abundant taxa with the highest mean relative abundance recorded in Lantana-based compost (77%); Fig 5B and Table 4. Class Holozoa was the most abundant non-fungal eukaryotic taxa across the composting days, with the highest values recorded on composting day 63 (mean relative abundance of 66%); Fig 5D and Table 4.

## Principal coordinate analysis (PCoA)

Bray-Curtis beta diversity of samples from different compost types and days showed distinct groupings (Fig 6A–6D). Notably, the percentage of variation in fungal community structure attributed to compost type and composting days is relatively small, with about 73% of the unexplained variance (Fig 6A, and 6C). The widest beta diversities for the fungal and non-fungal communities as influenced by compost treatments were observed in Tithonia-based and Grass-based composts, respectively. The least overlap and variation of diversity among the composting days in both fungal and non-fungal eukaryotic communities were observed on day 21 of composting, Fig 6C and 6D.

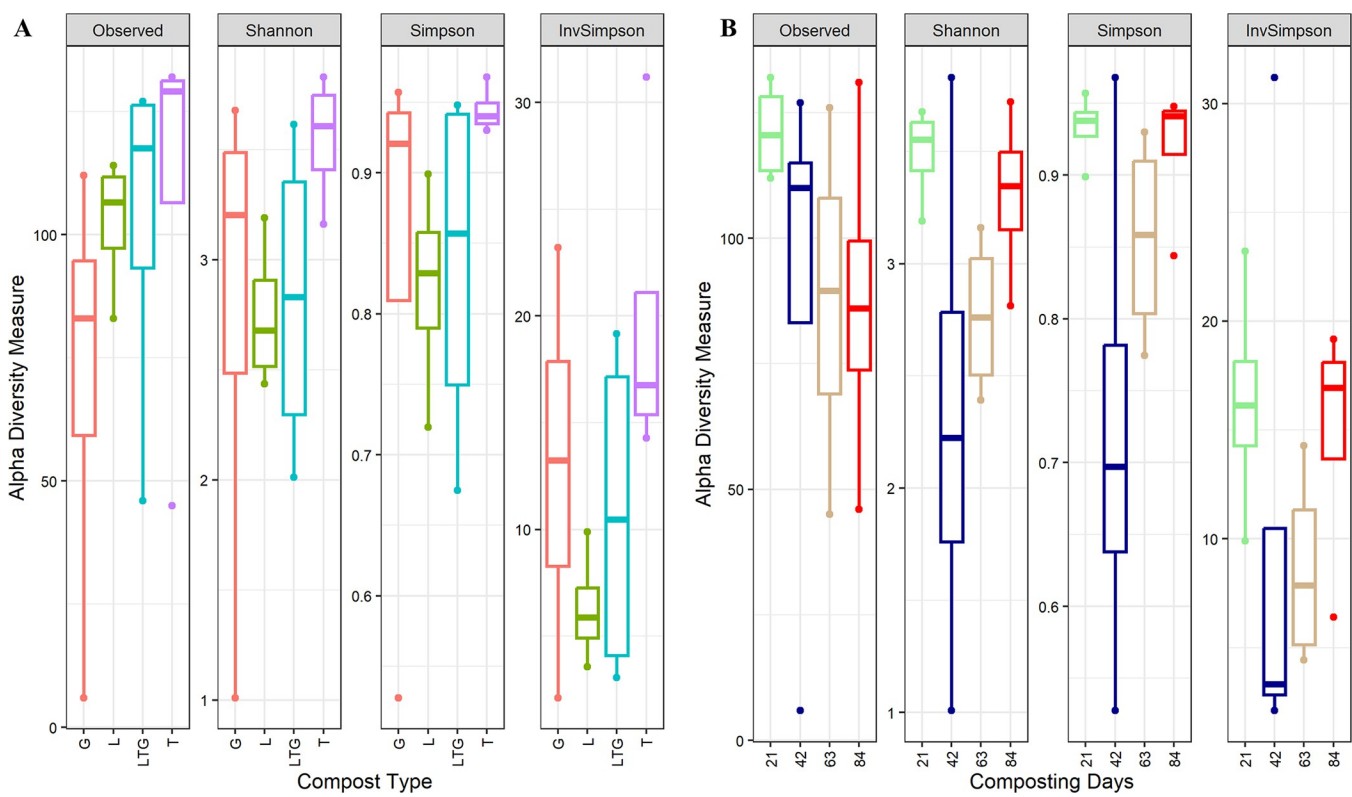

**Fig 4.** Alpha diversity metrics (Observed, Shannon, InvSimpson, and Simpson) of non-fungal eukaryotic communities under different composting treatments (A) and composting days (B). L, T, G, and LTG represent Lantana, Tithonia, Grass, and mixed (Lantana + Tithonia + Grass) based composts respectively. Day 21, day 42, day 63, and day 84 represent the effect of combined compost treatments at 21, 42, 63, and 84 days of composting.

**Table 3. Mean relative abundance (%) of the most abundant fungal eukaryotic classes in different compost treatments and composting days.**

| Class | Overall relative abundance | | Compost treatments | | | | Composting days | | | |
|---|---|---|---|---|---|---|---|---|---|---|
| | Mean | Standard deviation | G | L | LTG | T | 21 | 42 | 63 | 84 |
| Sordariomycetes | 61 | 18.3 | 64 | 56 | 57 | 68 | 67 | 74 | 47 | 60 |
| Dothideomycetes | 20 | 18.8 | 19 | 21 | 20 | 20 | 6 | 12 | 33 | 28 |
| Agaricomycetes | 8 | 7.4 | 9 | 8 | 9 | 6 | 18 | 4 | 6 | 3 |
| Eurotiomycetes | 4 | 2.4 | 3 | 4 | 7 | 3 | 4 | 4 | 5 | 4 |
| Mortierellomycetes | 2 | 3.3 | 0 | 3 | 3 | 1 | 0 | 3 | 4 | 1 |
| Saccharomycetes | 2 | 1.6 | 3 | 2 | 2 | 1 | 1 | 1 | 3 | 2 |
| Tremellomycetes | 1 | 3.1 | 0 | 4 | 1 | 0 | 3 | 0 | 0 | 1 |
| Pezizomycetes | 1 | 0.8 | 0 | 1 | 1 | 0 | 1 | 1 | 0 | 1 |
| Cystobasidiomycetes | 0 | 0.8 | 1 | 0 | 0 | 0 | 0 | 0 | 1 | 0 |
| Orbiliomycetes | 0 | 0.8 | 1 | 1 | 0 | 0 | 0 | 1 | 0 | 0 |
| Microbotryomycetes | 0 | 0.3 | 0 | 0 | 0 | 0 | 0 | 0 | 0 | 0 |
| Leotiomycetes | 0 | 0.2 | 0 | 0 | 0 | 0 | 0 | 0 | 0 | 0 |
| Rhizophlyctidomycetes | 0 | 0.2 | 0 | 0 | 0 | 0 | 0 | 0 | 0 | 0 |
| Laboulbeniomycetes | 0 | 0.1 | 0 | 0 | 0 | 0 | 0 | 0 | 0 | 0 |
| Ustilaginomycetes | 0 | 0 | 0 | 0 | 0 | 0 | 0 | 0 | 0 | 0 |
| Total | 100 | | 100 | 100 | 100 | 100 | 100 | 100 | 100 | 100 |

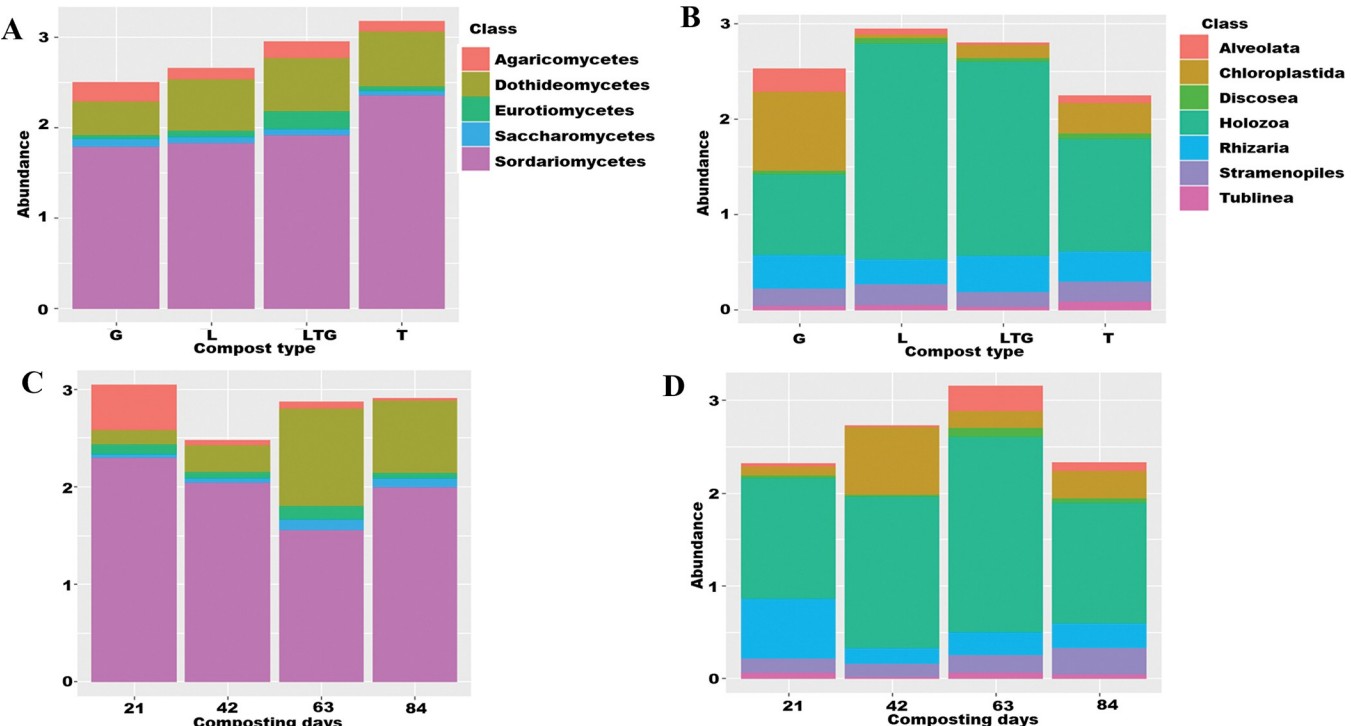

**Fig 5. A-D:** Relative abundances of fungal and non-fungal eukaryotic classes in various composting treatments (A, C respectively) and sampling days (B and D respectively). L, T, G, and LTG represent lantana, tithonia, grass, and mixed (lantana + tithonia + grass) based composts, respectively. 21, 42, 63, and 84 represent the effect of combined compost treatments at 21, 42, 63, and 84 days.

## Unique fungal and non-fungal eukaryotic taxa within compost environment

Venn diagrams to show the common OTUs and exclusive OTUs among different compost treatments and composting days were generated. Grass-based compost and mixed compost recorded the most unique core fungal taxa among the compost types, with three unique taxa for each of the two treatments (Fig 7A). Lantana-based compost had the highest non-fungal eukaryotic core taxa, recording seven unique taxa (Fig 7B). On the other hand, composting day 21 recorded the highest unique core fungal taxa among all the composting days, with 12 exclusive taxa (Fig 7C) while composting day 21 had the most unique non-fungal taxa (Fig 7D).

**Table 4. Mean relative abundance (%) of the most abundant non-fungal eukaryotic classes in different compost treatments and composting days.**

| Class | Overall relative abundance | | Compost treatments | | | | Days of composting | | | |
|---|---|---|---|---|---|---|---|---|---|---|
| | Mean | Standard deviation | G | L | LTG | T | 21 | 42 | 63 | 84 |
| Holozoa | 58 | 23.1 | 33 | 77 | 70 | 53.5 | 55 | 56 | 66 | 55 |
| Rhizaria | 13 | 11.8 | 16 | 9 | 15 | 13.5 | 28 | 6 | 8 | 12 |
| Chloroplastida | 13 | 24.5 | 33 | 1 | 6 | 14 | 4 | 30 | 7 | 13 |
| Stramenopiles | 8 | 4.8 | 7.8 | 7.5 | 6 | 10 | 7 | 6 | 6 | 12 |
| Alveolata | 4 | 6 | 8.5 | 2 | 1 | 3.5 | 2 | 1 | 9 | 4 |
| Discosea | 2 | 1.3 | 1 | 2 | 1.5 | 2.5 | 2 | 1 | 3 | 2 |
| Tubulinea | 2 | 1.4 | 2 | 1.5 | 1 | 3 | 3 | 1 | 2 | 2 |
| Total | 100 | | 100 | 100 | 100 | 100 | 100 | 100 | 100 | 100 |

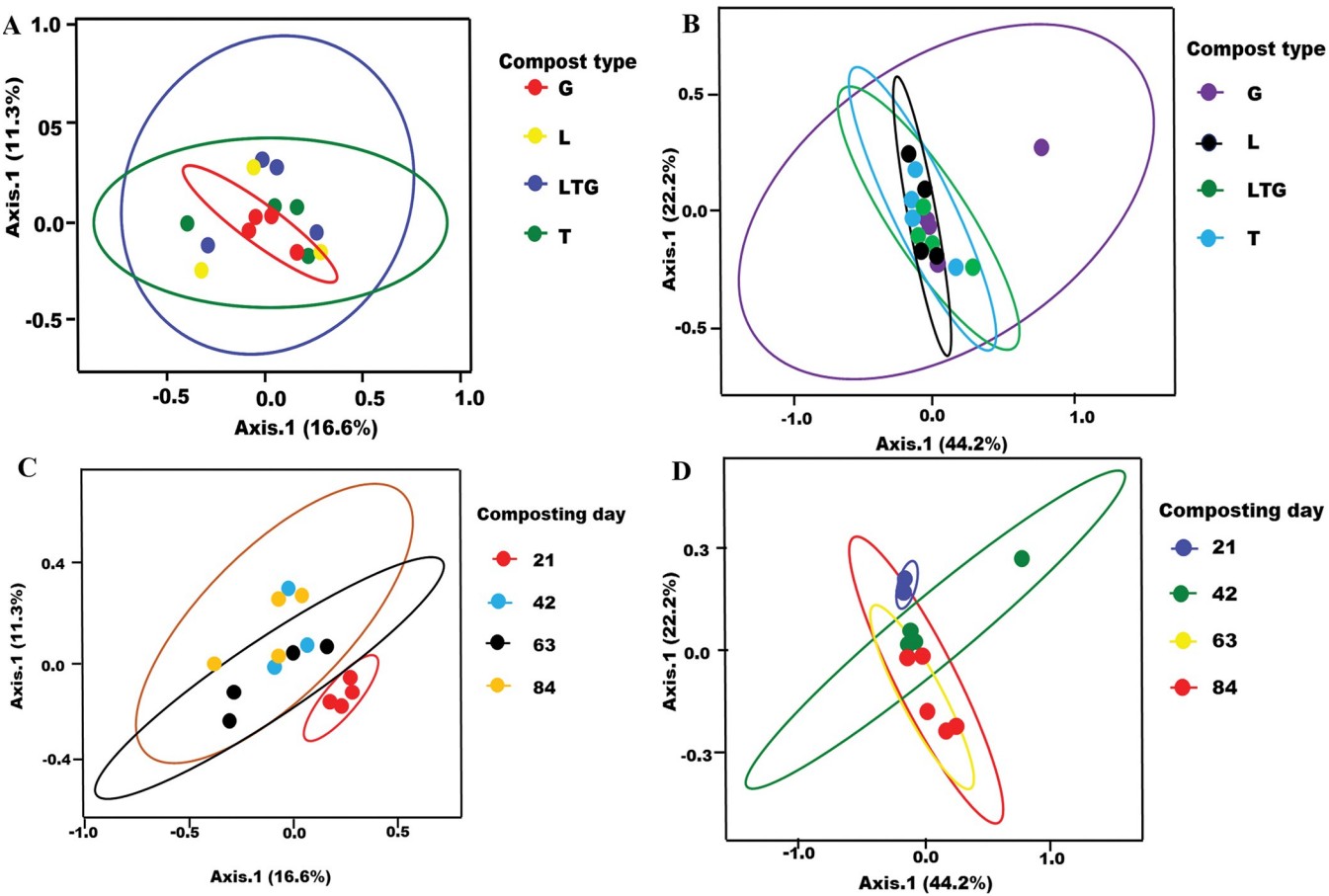

**Fig 6. A-D:** Principal coordinate analysis (PCoA) ordination plots based on the Bray-Curtis index at 95% confidence. The different compost groups are highlighted by ellipses.

### Fungal and non-fungal community interactions within the compost environment

Five (5) fungal classes (Agaricomycete, Eurotiomycetes, Dothideomycets, Saccharomycetes, and Sordariomycetes) were displayed to correlate with each other at different intensities (correlation values between 0.0 and 0.6) as shown in the interaction network. The class Sordariomycetes was the hub taxon contributing to the most network nodes, with 10 taxa belonging to the class. Consequently, this taxon had the most interactions with other fungal classes in the compost environment (Fig 8).

On the other hand, seven (7) non-fungal classes (Alveolata, Chloroplastida, Discosea, Holozoa, Rhizaria, Stramenopiles, and Tublinea) were shown to bear the most interactions within the compost environment. Class Holozoa had the most interactions with non-fungal eukaryotic taxa in the compost ecosystem (Fig 9). Taxa under this class supported most close interactions (mainly at a correlation of 0.2) within the non-fungal community; Fig 9.

### Physical-chemical drivers of compost fungal and non-fungal eukaryotic communities

Canonical correspondence analysis (CCA) ordination plots of environmental factors showed the significant (adj. p-value<0.01) influence of these factors on fungal and non-fungal eukaryotic biomes of compost (Fig 10A and 10B). Most fungal classes were responsive to less

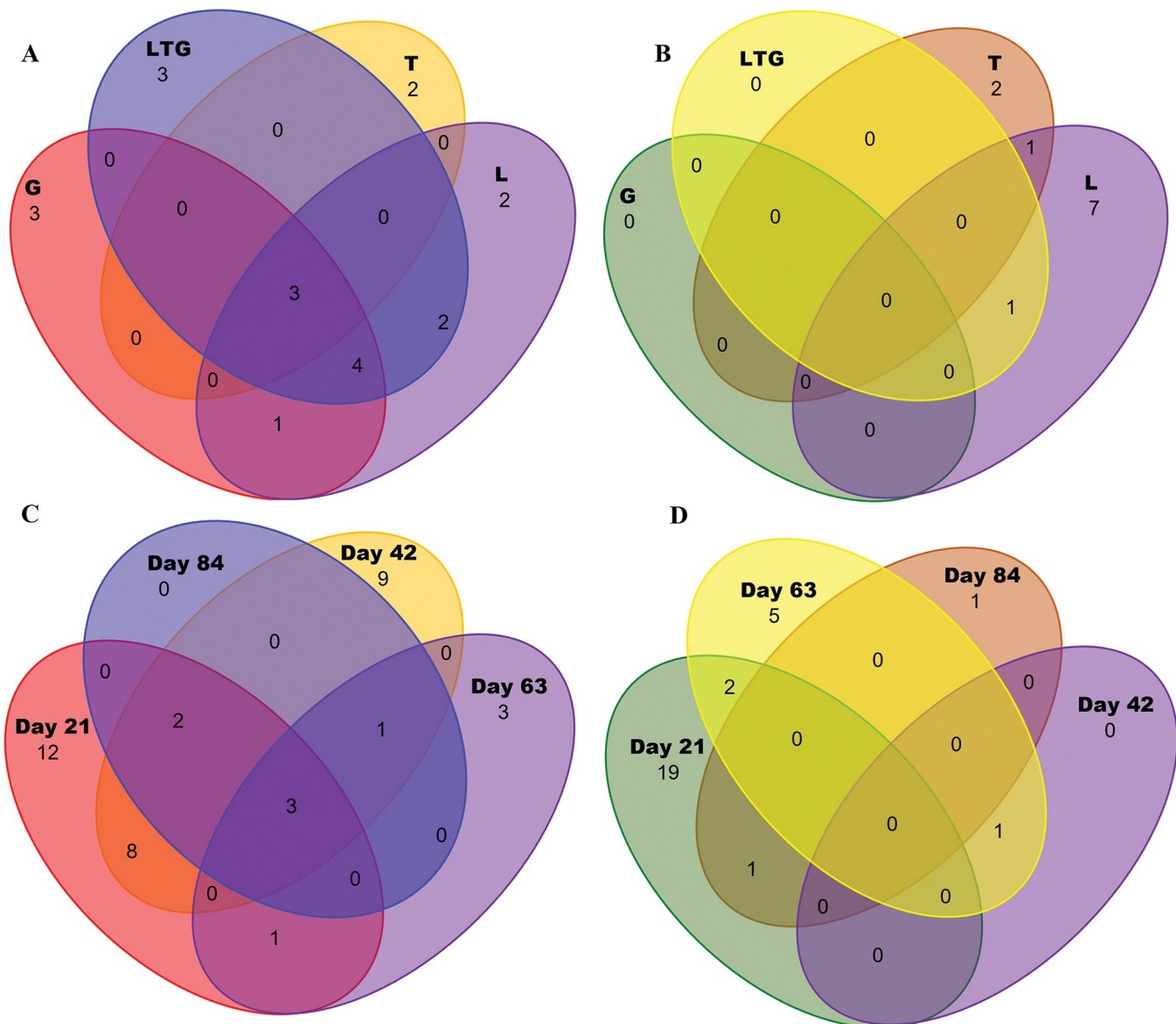

**Fig 7. A-D:** Venn diagram based on shared major core taxa of fungal and non-fungal eukaryotic communities under compost treatment (A and B), composting days (C and D).

ammonia, with Dothideomycetes and Laboulbeniomycetes having the and responded positively to a decrease in other physical-chemical states of compost. The fungal class Agaricomycetes was the most sensitive to Nitrates and Carbon. Non-fungal eukaryotic class Tublinea was uniquely responsive to decreasing temperature, Carbon, and Phosphorous, and increasing Nitrates. The frequency of class Discosea is associated with high pH, total nitrogen, and low nitrate levels.

## Discussion

Studying eukaryotic communities' structure in thermophilic composting is crucial to understanding the categories and succession of fungi that are useful for improved soil health and not

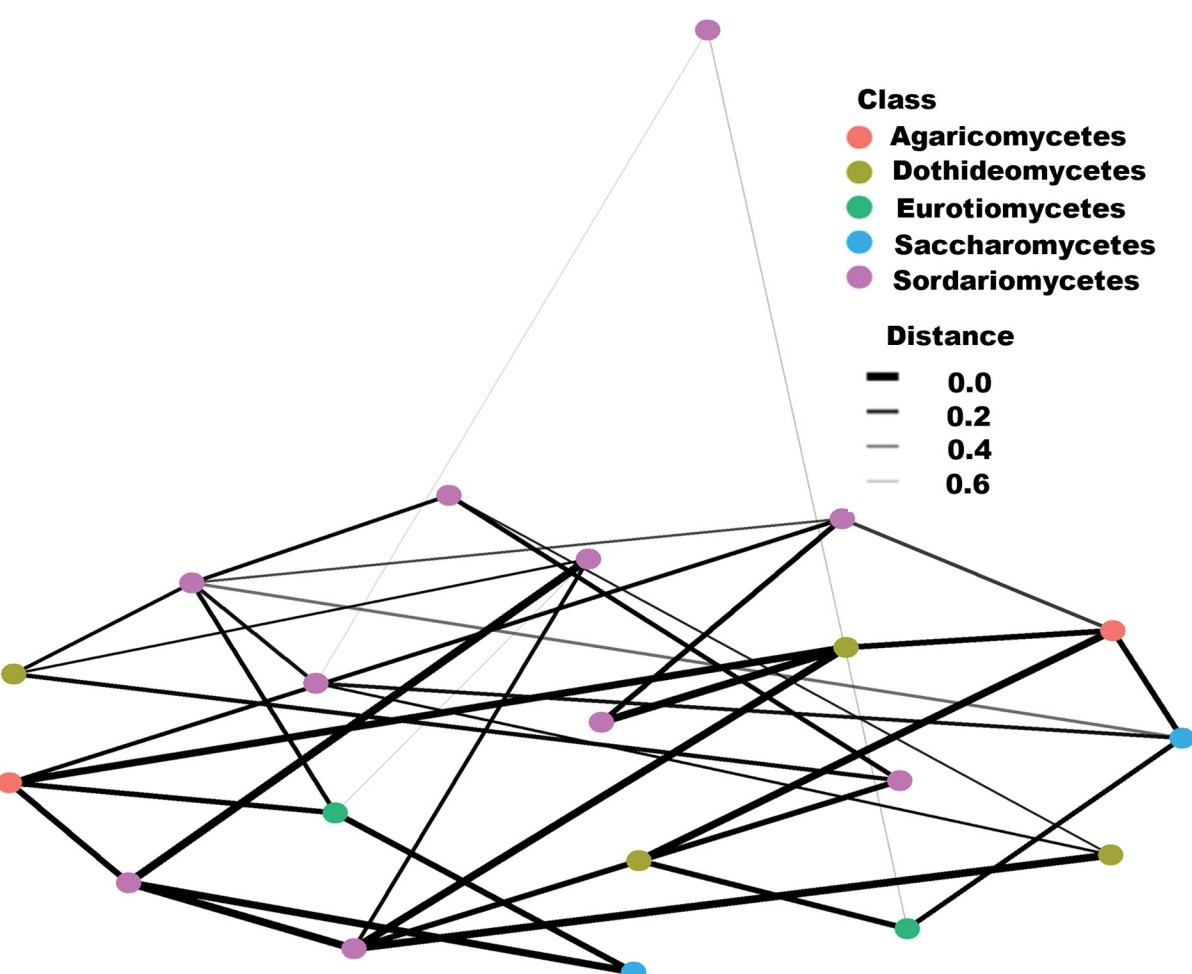

**Fig 8. Correlation network showing interactions among fungal eukaryotic classes driving composting.** The network nodes represent genera, whereas the edges represent microbe-microbe interaction weights. Networks were constructed based on the top 20 classes. Various color codes in the networks represent labeled genera within the classes listed on the grid.

harmful to humans, plants, and environmental health [28]. This knowledge is also worthwhile in the optimization of compost quality standards for safer crop production [29].

Physical-chemical conditions are indicators of the humification rate that ultimately brings biological and nutritional stability to compost [30]. Humification is preceded by rapid biogeo-chemical phases which are microbially driven, breaking down complex polymers into organic acids [31]. The high variability within Tithonia-based compost can be attributed to the signifi-cant physical-chemical changes within this treatment compared to other treatments along the composting process. The strong positive correlation between factors such as Carbon and tem-perature points to the co-dependence of these two elements in the physical-chemical nature of compost. The breakdown of Carbon by microbes leads to a temperature increase in the com-post environment [1]. This temperature increase is responsible for nutrient mineralization and, ultimately humification of compost [30].

The high fungal biodiversity (richness) at 84 days of composting compared to preceding composting days implies that more extended composting periods are necessary for compost stability [32]. Several authors have reported the recruitment of more fungal categories respon-sible for the maturation and, ultimately, humification of compost [33–35]. The diversity of the

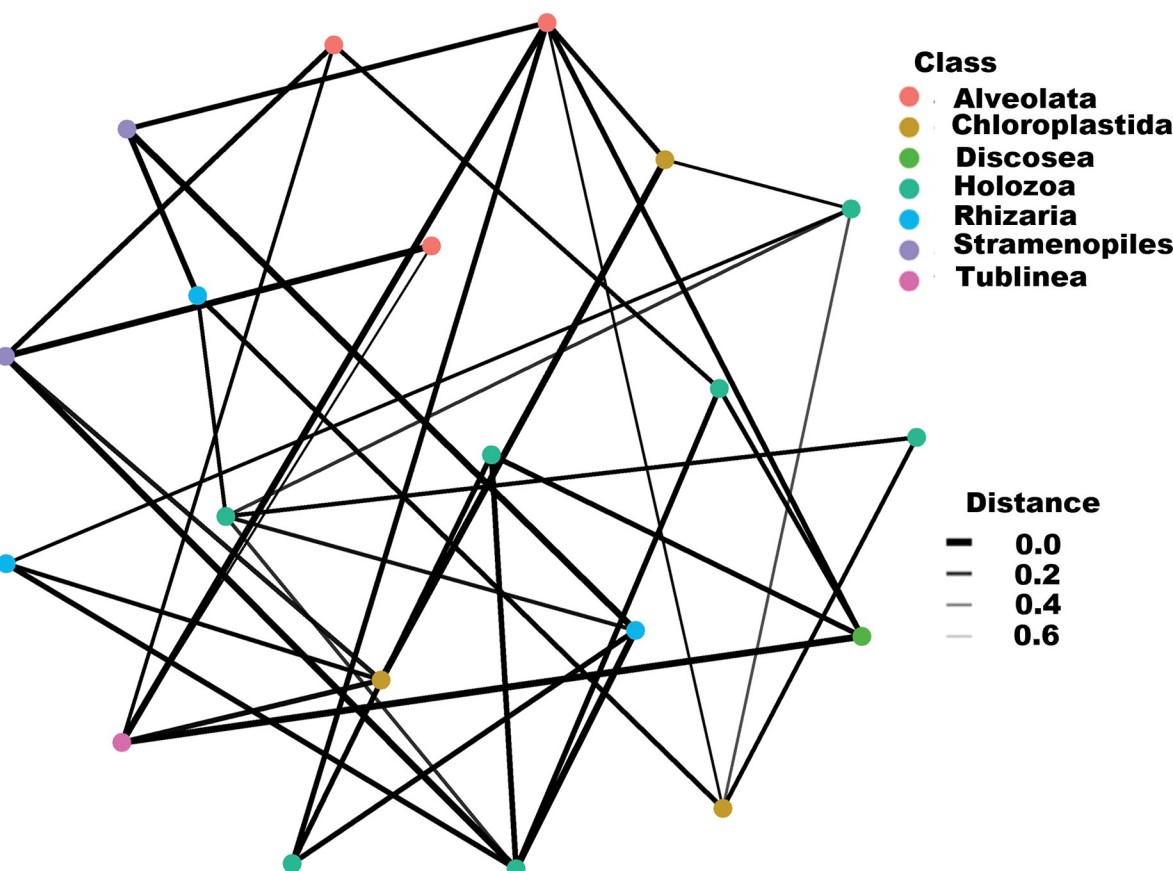

**Fig 9. Correlation network showing interactions among non-fungal eukaryotic classes driving composting.** The network nodes represent genera, whereas the edges represent microbe-microbe interaction weights. Networks were constructed based on the top 20 classes. Various color codes in the networks represent labeled genera within the classes listed on the grid.

fungal community in Lantana-based compost is attributable to the complexity of Lantana compared to other materials and hence ecosystem recruitment of diverse fungal categories with the capacity to metabolize this material.

The ubiquitous fungi taxa in all composting treatments and days (Sordariomycetes, Agaricomycetes, Dothidiomycetes, Eurotiomycetes, and Saccharomycetes) were reported as resident classes in compost [36–38]. The dominance of class Sordariomycetes in the compost environment indicates its critical role in the evolution of compost and persistence in the compost ecosystem. This class has been reported as having a superior metabolic capability to other fungal classes [39–41].

Classes Alveolata, Chloroplastida, Discosea, Holozoa, Rhizaria, Stramenopiles, and Tublinea have been reported as resident non-fungal eukaryotes that colonize organic wastes [42]. These eukaryotes have been reported as active soil protists [43] with metabolic functions that are essential nutrient cycling pathways. The presence of non-fungal eukaryotic class Chloroplastida in all compost treatments and days affirms the suitability of compost as a soil health input bringing in novel microbes with the ability to improve soil physical-chemical state. Algal biota such as Chloroplastida has been reported as having broad metabolic capabilities in fixing nitrogen, acting as plant growth promotion and disease control [44].

Fungal class Agaricomycetes has been significantly associated with Carbon and Nitrogen cycling in the ecosystem [10] hence their dominant association with nitrates and Carbon

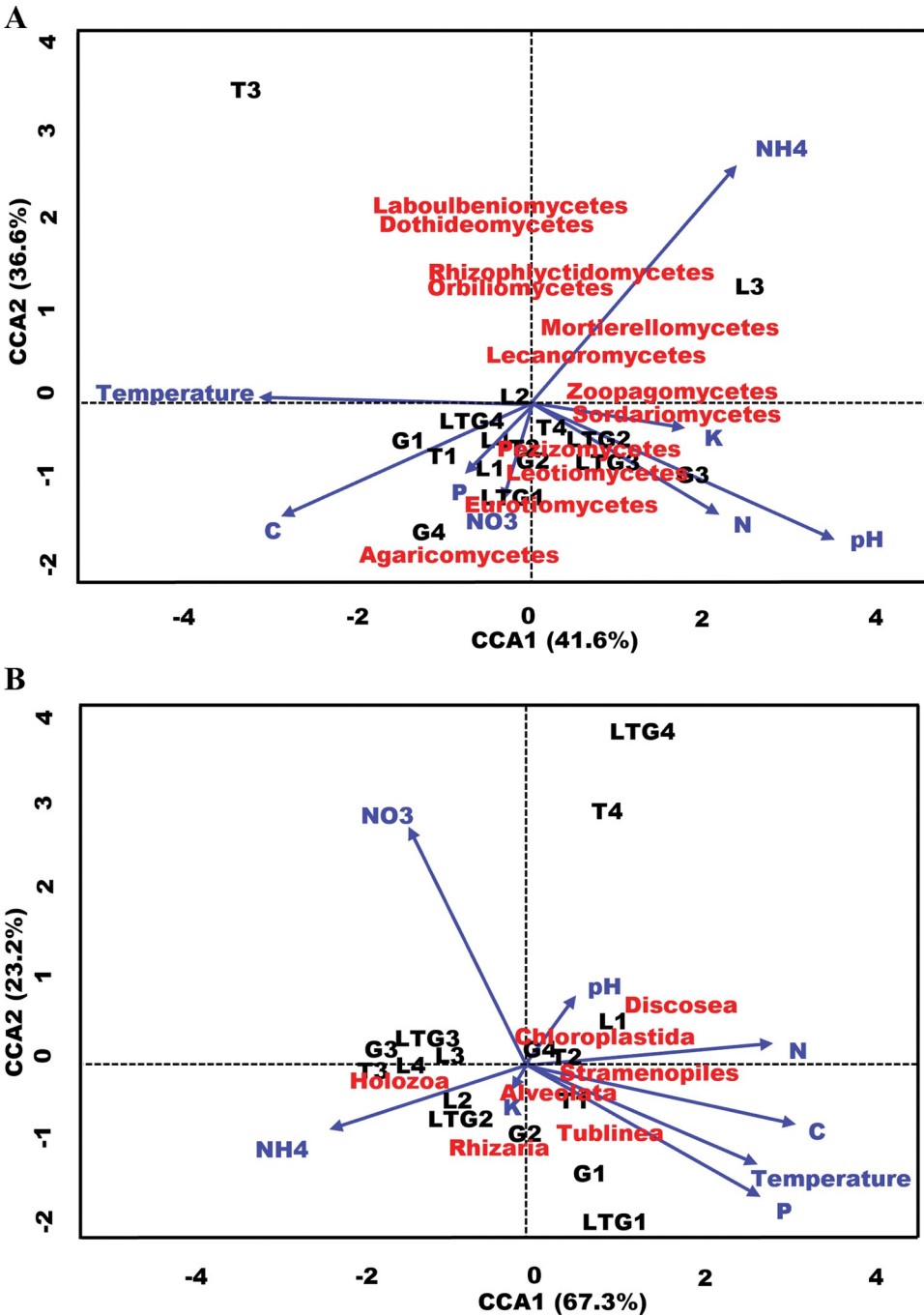

**Fig 10. A, B:** Canonical correspondence analysis (CCA) plots showing the effect of explanatory physical-chemical variables on the different fungal (A) and non-fungal (B) microbiomes using a significance threshold of 0.05. C-Organic Carbon, N- Total Nitrogen, K- Potassium, NO3- Nitrates, NH4- Ammonia.

during composting. This primes this class as potential soil Carbon and nitrogen cycling microbial category as a suitable candidate for soil improvement inoculants.

The change of unique fungal ASVs in the different composting days implies a clear evolution of fungal communities as influenced by changing nature of materials in compost as well as physical-chemical parameters and vice-versa [28, 36, 45]. The high unique core taxa in

Lantana-based compost points to the role of the eukaryotes in the unique degradation of material in this composting treatment. Lantana has been reported as inhibitory to some classes of microorganisms [46–48], therefore requires specialized classes of microbes to break down.

Hub taxa have a direct inhibitory or facilitative role in the proliferation and survival of other microbes affecting overall interconnected communities [49, 50]. The genera in the class Sordariomycetes which is the major fungal hub taxon has been reported as the first line of colonizers of recalcitrant material in soil, therefore breaking this material into simpler forms available for other fungal and prokaryotic communities [39, 41]. The degradation of complex material such as lignocellulose in ecosystems like compost is driven by a synergistic action of oxidative and hydrolytic enzymes that break the linkages within the material [51]. This process requires a variety of interactions among different categories of microorganisms [52].

## Conclusion

This study demonstrates how various sources of green material and composting days affect the evolution of diverse fungal and non-fungal eukaryotic communities. Fungal and non-fungal eukaryotic diversity and abundance changed significantly with compost type and composting days. High throughput ITS and 18S gene sequencing indicated that the dominant classes during composting included Sordariomycetes, Agaricomycetes, Dothidiomycetes, Eurotiomycetes, and Saccharomycetes of fungi. On the other hand, Alveolata, Chloroplastida, Discosea, Holozoa, Rhizaria, Stramenopiles, and Tublinea were the dominant non-fungal eukaryotic classes. The highest abundance and major contribution to interactions by Sordariomycetes and Holozoa in the fungal and non-fungal eukaryotic communities respectively assert their contribution in the respective communities in the compost environment. The findings of this study extend the understanding of the evolution of fungal and non-fungal eukaryotes as well as key players in communal interactions during composting.

## Acknowledgments

The authors wish to thank the Pan African Network for Bioinformatics Training (H3ABioNet) through Caleb Kibet and Andrew Espira of the International Centre of Insect Physiology and Ecology (ICIPE) node for providing necessary computational infrastructure and invaluable support to bring this project to a good end. The authors acknowledge the support received from SysCom Kenya research assistant James Karanja during fieldwork.

## Author Contributions

**Conceptualization:** Felix Matheri, Anne Kelly Kambura, Maina Mwangi, Edward Karanja, Noah Adamtey, Edwin Mwangi, Steven Runo.

**Data curation:** Felix Matheri, Anne Kelly Kambura, Maina Mwangi, Edward Karanja, Noah Adamtey, Kennedy Wanjau.

**Formal analysis:** Felix Matheri, Anne Kelly Kambura, Kennedy Wanjau, Edwin Mwangi, Steven Runo.

**Funding acquisition:** Felix Matheri, Maina Mwangi, Edward Karanja, Noah Adamtey, Chrysantus Mbi Tanga, David Bautze, Steven Runo.

**Investigation:** Felix Matheri, Anne Kelly Kambura, Maina Mwangi, Edward Karanja, Noah Adamtey, Edwin Mwangi, Chrysantus Mbi Tanga, David Bautze, Steven Runo.

**Methodology:** Felix Matheri, Anne Kelly Kambura, Maina Mwangi, Edward Karanja, Noah Adamtey, Edwin Mwangi, Chrysantus Mbi Tanga, Steven Runo.

**Project administration:** Felix Matheri, Maina Mwangi, Edward Karanja, Noah Adamtey, Chrysantus Mbi Tanga, Steven Runo.

**Resources:** Felix Matheri, Maina Mwangi, Edward Karanja, Noah Adamtey, Chrysantus Mbi Tanga, David Bautze, Steven Runo.

**Software:** Felix Matheri, Kennedy Wanjau, Chrysantus Mbi Tanga, David Bautze.

**Supervision:** Anne Kelly Kambura, Maina Mwangi, Noah Adamtey, Steven Runo.

**Validation:** Felix Matheri, Anne Kelly Kambura, Maina Mwangi, Edward Karanja, Noah Adamtey, Kennedy Wanjau, Edwin Mwangi, Chrysantus Mbi Tanga, Steven Runo.

**Visualization:** Felix Matheri, Anne Kelly Kambura, Kennedy Wanjau, Chrysantus Mbi Tanga, David Bautze.

**Writing – original draft:** Felix Matheri, Anne Kelly Kambura, Edward Karanja, Noah Adamtey, Edwin Mwangi, Chrysantus Mbi Tanga.

**Writing – review & editing:** Felix Matheri, Anne Kelly Kambura, Maina Mwangi, Edward Karanja, Noah Adamtey, Kennedy Wanjau, Edwin Mwangi, Chrysantus Mbi Tanga, David Bautze, Steven Runo.

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
