## [Decision Letter · Decision Letter 0]

16 Jan 2023

PONE-D-22-30019Evolution of fungal and non-fungal eukaryotic communities in response to thermophilic co-composting of various nitrogen-rich green feedstocks.PLOS ONE

Dear Dr. Matheri,

Thank you for submitting your manuscript to PLOS ONE. After careful consideration, we feel that it has merit but does not fully meet PLOS ONE’s publication criteria as it currently stands. Therefore, we invite you to submit a revised version of the manuscript that addresses the points raised during the review process. Kindly check the manuscript thoroughly to improve its readability and to correct the grammatical mistakes. Moreover, also highlight the novelty of the work and research gaps the study has addressed. Please submit your revised manuscript by Mar 02 2023 11:59PM. If you will need more time than this to complete your revisions, please reply to this message or contact the journal office at plosone@plos.org. Please include the following items when submitting your revised manuscript:

We look forward to receiving your revised manuscript.

Kind regards,

Sudeshna Bhattacharjya, Ph.D

Academic Editor

PLOS ONE

Journal Requirements:

When submitting your revision, we need you to address these additional requirements. 1. Please ensure that your manuscript meets PLOS ONE's style requirements, including those for file naming. The PLOS ONE style templates can be found at https://journals.plos.org/plosone/s/file?id=wjVg/PLOSOne_formatting_sample_main_body.pdf and https://journals.plos.org/plosone/s/file?id=ba62/PLOSOne_formatting_sample_title_authors_affiliations.pdf 2. We note that the grant information you provided in the ‘Funding Information’ and ‘Financial Disclosure’ sections do not match.  When you resubmit, please ensure that you provide the correct grant numbers for the awards you received for your study in the ‘Funding Information’ section. 3. Thank you for stating the following financial disclosure: "This research received partial financial support from Biovision Foundation, Coop Sustainability Fund, Liechtenstein Development Service (LED), Swiss Agency for Development and Cooperation (SDC)  through the Research Institute of Organic Agriculture (FiBL) and the International Centre for Insect Physiology and Ecology (Grant No: 1040). The funders had no role in study design, data collection and analysis, decision to publish, or preparation of the manuscript. Therefore, the views expressed herein do not necessarily reflect the official opinion of the donors." 
Please state what role the funders took in the study.  If the funders had no role, please state: "The funders had no role in study design, data collection and analysis, decision to publish, or preparation of the manuscript." If this statement is not correct you must amend it as needed. Please include this amended Role of Funder statement in your cover letter; we will change the online submission form on your behalf. 

Reviewers' comments:

Reviewer's Responses to Questions

**Comments to the Author**

1. Is the manuscript technically sound, and do the data support the conclusions?

Reviewer #1: Partly

Reviewer #2: Partly

2. Has the statistical analysis been performed appropriately and rigorously? 

Reviewer #1: Yes

Reviewer #2: Yes

3. Have the authors made all data underlying the findings in their manuscript fully available?

Reviewer #1: Yes

Reviewer #2: Yes

4. Is the manuscript presented in an intelligible fashion and written in standard English?

Reviewer #1: No

Reviewer #2: Yes

5. Review Comments to the Author

Reviewer #1: Page 16 line 190, 191 and 290: during composting, we know that the carbon content is decreasing throughout the composting, while nitrogen and phosphorus are increasing, while the temperature is increasing from the beginning until the maturation phase, I think it is contradictory to say that there is a positive correlation between temperature and carbon. so how could the authors explain the positive correlation between temperature and carbon?

Page 16 line 202, 204 and 206 : what do the authors mean by and 84 days of composting ??? it doesn't make sense ? the authors should change these sentences and review the whole document ?

Page 18 line 243 and 253; are these a title or what?

Conclusion: authors should improve the conclusion to be more attractive.

English should be improved.

I think this article is lacking in novelty.

The quality of all figures is poor and unreadable, the authors should improve the quality of all figures.

Reviewer #2: The present manuscript investigates the structure and contribution of fungal and non-fungal community during thermophilic composting, through the implementation of four experimental schemes (Lantana, Tithonia, Grass, mixture). The research is in general interesting. The abstract section contains information which belong to the materials methods. Lines 30-35 should not include methodology but the actual findings of the research. Authors state that findings of this study unravel influence composting materials on the eukaryotic community structure but this is not supported in the previous lines (1-41). The introduction section is very general; it includes many references but very little information on the actual subject. In lines 70-72 is reported that Lantana camara and Tithonia diversifolia are phytochemicaly complex and the Lantana contain complex polymers without further details. Authors should consider the structure and the content of the introduction, which should increase and contain useful and specific scientific information. Please clearly report (using a plain table) the physicochemical characteristics of the compost schemes. All figures have poor resolution and should be resized. Lines 188-192 are more like a general statement and not a detailed presentation of the results, where no correlation is reported and the same apply for all the section of the results. Please include the relative abundance of the main fungal classes and please do the same for the non-fungal communities. Lines 245-249, Lines 256-258 are not presenting any results. Lines 271-271 are confusing. What the authors mean by ‘’preferred less ammonia and responded positively to decrease in other....states of the compost’’? There is no discussion on PCA and on the statistical analysis. The conclusion section is very general.

6. PLOS authors have the option to publish the peer review history of their article (what does this mean?). If published, this will include your full peer review and any attached files.

Reviewer #1: **Yes: **Saloua Biyada

Reviewer #2: No

---

## [Author Response · Author response to Decision Letter 0]

27 Feb 2023

I would like to appreciate the opportunity to submit a revised version of the manuscript entitled “Evolution of fungal and non-fungal eukaryotic communities in response to thermophilic co-composting of various nitrogen-rich green feedstocks” to be considered for publication in your Journal as an original article. Kindly also note that the referred line numbers are on the “Revised Manuscript with Track Changes” files.

We highly appreciate and welcome your comments and those of the reviewers and have addressed the points raised during the review process as follows;

Reviewer Comments:

Reviewer #1: 

Reviewer comment No. 1: Page 16 line 190, 191 and 290: during composting, we know that the carbon content is decreasing throughout the composting, while nitrogen and phosphorus are increasing, while the temperature is increasing from the beginning until the maturation phase, I think it is contradictory to say that there is a positive correlation between temperature and carbon. so how could the authors explain the positive correlation between temperature and carbon? 

Authors’ response: 

The two variables tended to increase and decrease together, thus making the correlation value positive. It is true that during composting, carbon decreases. However, temperature also decreases during the composting period. Of all the sampling days, the highest temperature and carbon were recorded at 21 days of composting, with both variables decreasing along the composting periods. Plausible reasons for the positive correlation of the two variables are in lines 368 to 372. The authors have enriched this manuscript by adding table 2 and lines 218-221 containing data on the trends of physical-chemical variables along the composting process to provide more information.

Reviewer comment No. 2: Page 16 line 202, 204 and 206: what do the authors mean by and 84 days of composting ??? it doesn't make sense ? the authors should change these sentences and review the whole document ? 

Authors’ response:

This statement was used to indicate the number of days (sampling times/points) from the time of heaping compost to maturation within 84 days. Nonetheless, the word has been rephrased to read as “day 84 of composting” to qualify the statements throughout the manuscript.

Reviewer comment No. 3: Page 18 line 243 and 253; are these a title or what?

Authors’ response:

The titles have been revised on page 16 lines 306-308 and 320-322 respectively to read as “Unique fungal and non-fungal eukaryotic taxa within compost environment” and “Fungal and non-fungal community interactions within compost environment.”

Reviewer comment No. 4: Conclusion: authors should improve the conclusion to be more attractive. 

Authors’ response:

The conclusion has been rewritten as advised.

Reviewer comment No. 5: English should be improved 

Authors’ response:

The authors have read through the manuscript and where applicable, the most appropriate adjectives have been adopted to improve the manuscript as recommended.

Reviewer comment No. 6: I think this article is lacking in novelty

Authors’ response:

The authors believe that the novelty of this article lies in lines 56-59 , 71-76 and 85-88. Lines 64-70, and 90-96 have been added to provide more information on the novelty of the study.

Reviewer comment No. 7: The quality of all figures is poor and unreadable, the authors should improve the quality of all figures

Authors’ response:

The figures have been improved for better resolution.

Reviewer 2

Reviewer comment No. 1: The present manuscript investigates the structure and contribution of fungal and non-fungal community during thermophilic composting, through the implementation of four experimental schemes (Lantana, Tithonia, Grass, mixture). The research is in general interesting. The abstract section contains information which belong to the materials methods. Lines 30-35 should not include methodology but the actual findings of the research.

Authors’ response: 

The abstract has been revised in lines 31-39 to describe the main objective(s) of the study, explain how the study was done, and a summary of the most important results and their significance. Methodological details have been removed as advised.

Reviewer comment No. 2: Authors state that findings of this study unravel influence composting materials on the eukaryotic community structure but this is not supported in the previous lines (1-41).

Authors’ response: 

The influence of composting material on eukaryotic community structure is shown in lines 29-31 and results in lines 36-44.

Reviewer comment No. 3: The introduction section is very general; it includes many references but very little information on the actual subject.

Authors’ response: 

More specific information on the influence of various feedstocks on eukaryotic communities has been added and appropriately cited in lines 64-96 within the manuscript.

Reviewer comment No. 4: In lines, 70-72 is reported that Lantana camara and Tithonia diversifolia are phytochemicaly complex and the Lantana contain complex polymers without further details.

Authors’ response: 

Further details regarding the phytochemical complexities of Lantana and Tithonia have been added in lines 82 to 84 within the manuscript.

Reviewer comment No. 5: Authors should consider the structure and the content of the introduction, which should increase and contain useful and specific scientific information. 

Authors’ response: 

More specific information on the influence of various feedstocks on eukaryotic communities has been added and appropriately cited in lines 64-96 within the manuscript.

Reviewer comment No. 6: Please clearly report (using a plain table) the physicochemical characteristics of the compost schemes.

Authors’ response:

Table 2 and lines 218-221 have been added to the results section for clarity. 

Reviewer comment No. 7: All figures have poor resolution and should be resized.

Authors’ response:

The figures have been improved for better resolution.

Reviewer comment No. 8: Lines 188-192 are more like a general statement and not a detailed presentation of the results, where no correlation is reported and the same apply for all the section of the results.

Authors’ response:

Detailed results (tables 2) and explanations have been added in lines 218-234 to enhance the manuscript 

Reviewer comment No. 9: Please include the relative abundance of the main fungal classes and please do the same for the non-fungal communities.

Authors’ response:

This has been added in text lines 271-279 and tables 3A and 3B to provide the computed relative abundances to the manuscript.

Reviewer comment No. 10: Lines 245-249, Lines 256-258 are not presenting any results.

Authors’ response:

Lines 308-315 and 322-332 have been added to improve the result presented.

Reviewer comment No. 11: Lines 271-271 are confusing.

Authors’ response:

The lines (345-346) have been rephrased for clarity.

Reviewer comment No. 12: What the authors mean by ‘’preferred less ammonia and responded positively to decrease in other....states of the compost’’?.

Authors’ response:

This implies that they thrive under lower levels of ammonia. Nonetheless, the line/s (345-346) has been rephrased for clarity.

Reviewer comment No. 13: there is no discussion on PCA and on the statistical analysis

Authors’ response:

Discussion on PCA has been added to lines 366-368.

Reviewer comment No. 14: The conclusion section is very general

Authors’ response:

The conclusion has been rewritten as advised.

We hope that we have adequately responded to all the comments raised regarding this manuscript in readiness for its publication in your journal.

---

## [Decision Letter · Decision Letter 1]

5 Apr 2023

PONE-D-22-30019R1Evolution of fungal and non-fungal eukaryotic communities in response to thermophilic co-composting of various nitrogen-rich green feedstocksPLOS ONE

Dear Dr. Felix Matheri,

Thank you for submitting your manuscript to PLOS ONE. Reviewers have submitted their comments and we are positive about the manuscript. However, the manuscript still needs some minor revisions before it can be processed further. Therefore, we invite you to submit a revised version of the manuscript that addresses the points raised during the review process.

We look forward to receiving your revised manuscript.

Kind regards,

Sudeshna Bhattacharjya, Ph.D

Academic Editor

PLOS ONE

Journal Requirements:

Reviewers' comments:

Reviewer's Responses to Questions

**Comments to the Author**

1. If the authors have adequately addressed your comments raised in a previous round of review and you feel that this manuscript is now acceptable for publication, you may indicate that here to bypass the “Comments to the Author” section, enter your conflict of interest statement in the “Confidential to Editor” section, and submit your "Accept" recommendation.

Reviewer #1: (No Response)

Reviewer #2: All comments have been addressed

2. Is the manuscript technically sound, and do the data support the conclusions?

Reviewer #1: (No Response)

Reviewer #2: Yes

3. Has the statistical analysis been performed appropriately and rigorously? 

Reviewer #1: (No Response)

Reviewer #2: Yes

4. Have the authors made all data underlying the findings in their manuscript fully available?

Reviewer #1: (No Response)

Reviewer #2: Yes

5. Is the manuscript presented in an intelligible fashion and written in standard English?

Reviewer #1: (No Response)

Reviewer #2: Yes

6. Review Comments to the Author

Reviewer #1: Thank you very much for the authors efforts to respond to my comments, but in the still you should improve your manuscript, there are some further comments.

line 75: Please correct "in invitro" by dropping "in".

Line 113,145: write "icipe" in capital letters.

Table 2: Authors should indicate the meaning of each symbol and the unit: for example, Carbon (C), ditto for all others (N, P .....).

line 261: regarding the fungus class nomenclature, the authors should remove the italic character style. please review the overall manuscript

You still need to demonstrate the novelty of your manuscript, it is not clear.

Reviewer #2: (No Response)

7. PLOS authors have the option to publish the peer review history of their article (what does this mean?). If published, this will include your full peer review and any attached files.

Reviewer #1: **Yes: **Saloua Biyada

Reviewer #2: No

---

## [Author Response · Author response to Decision Letter 1]

26 Apr 2023

I would like to appreciate the opportunity to submit a revised version of the manuscript entitled “Evolution of fungal and non-fungal eukaryotic communities in response to thermophilic co-composting of various nitrogen-rich green feedstocks” to be considered for publication in your Journal as an original article. Kindly also note that the referred line numbers are on the “Revised Manuscript with Track Changes” file.

We highly appreciate and welcome your comments and those of the reviewers and have addressed the points raised during the review process as follows;

Reviewer Comments:

Reviewer #1: 

Thank you very much for the authors efforts to respond to my comments, but in the still you should improve your manuscript, there are some further comments.

Reviewer comment No. 1: line 75: Please correct "in invitro" by dropping "in".

Authors’ response: 

The word ‘in’ has been dropped and replaced with ‘under’.

Reviewer comment No. 2: Line 113,145: write "icipe" in capital letters.

Authors’ response:

This is acknowledged and changed throughout the manuscript.

Reviewer comment No. 3: Table 2: Authors should indicate the meaning of each symbol and the unit: for example, Carbon (C), ditto for all others (N, P .....).

Authors’ response:

This has been corrected as advised.

Reviewer comment No. 4: line 261: regarding the fungus class nomenclature, the authors should remove the italic character style. please review the overall manuscript.

Authors’ response:

This has been changed throughout the text as advised.

Reviewer comment No. 5: You still need to demonstrate the novelty of your manuscript, it is not clear.

Authors’ response:

The authors aver that the novelty of the study is on lines 25-29, 72-80. The rationale for this study is articulated within the manuscript in lines 93-101, where the justification for studying the diverse green materials has been presented “These materials have, however, been reported to have antimicrobial activity in in-vitro experiments. Furthermore, the phytochemical analysis has shown differences in their complexities, thus possibly requiring various periods to break down. Therefore, it is necessary to understand these materials' influence on the biological and physical-chemical stability of compost” and lines 65-75 “Some of these materials have been reported under invitro studies as having inhibitory properties on fungal and non-fungal organisms [13, 14]. The phytochemical complexities of Lantana camara and Tithonia diversifolia have been previously studied. Lantana camara has been reported to contain more complex polymers and thus possibly requires more microbial categories to break down into agriculturally useful material. For example, the phytochemical composition analysis of L. camara showed 23.3% crude fiber, 26% cellulose, 16.2% lignin, and 21% hemicellulose, while T. diversifolia contains 11.2%, 17%, 7%, and 16% of these elements respectively [15, 16, 17]. Furthermore, the different complexities of the composting material could require assorted composting times to produce a biogeochemically stable product. It is, therefore, necessary to evaluate the influence of composting time on the biological and physical-chemical quality of compost”

The importance of Eukaryotes in the ecosystems and their limited study is enumerated in lines 51-56 “Though less studied, eukaryotes are an important microbial category in the ecosystem with diverse feeding and genetic guilds [6, 7]. Eukaryotes degrade recalcitrant material such as Carbon-rich polymers, thus contributing to nutrient cycling in the compost environment [8]. The complex structure and function of a large number of compost microorganisms are directly affected by various environmental factors such as temperature, moisture, Carbon/Nitrogen ratio, and pH, among others”. 

The use of next-gen sequencing to study eukaryotes is justified in lines 124-130 “Numerous studies have assessed the microbial prokaryotic community in the composting process using culture-dependent and culture-independent methods [2, 4, 6, 10]. However, there is still a limited understanding of eukaryotic communities’ structure in the composting process, concerning the composition and complexity of the composting materials especially utilizing high-throughput sequencing technology. The effect of the Lantana camara and Tithonia diversifolia on microbial community structure in complex ecosystems such as the compost environment has not also been done, despite their widespread adoption by farmers as soil nutrient amendment materials”.

Moreover, lines 29-30, and 57-59 have been added to increase clarity.

Reviewer 2

Reviewer #2: (No Response)

Authors’ response:

The authors appreciate the great input from the reviewer.

We reiterate our appreciation to the reviewers for their kind comments which have helped improve the manuscript and hope that we have adequately responded to all the comments raised regarding this manuscript in readiness for its publication in your journal. 

Yours Sincerely 

Felix Matheri

Department of Biochemistry, Microbiology, and Biotechnology, Kenyatta University, 

P.O Box 30772-00100 Nairobi.

---

## [Decision Letter · Decision Letter 2]

15 May 2023

Evolution of fungal and non-fungal eukaryotic communities in response to thermophilic co-composting of various nitrogen-rich green feedstocks

PONE-D-22-30019R2

Dear Dr. Matheri,

We’re pleased to inform you that your manuscript has been judged scientifically suitable for publication and will be formally accepted for publication once it meets all outstanding technical requirements.

Kind regards,

Sudeshna Bhattacharjya, Ph.D

Academic Editor

PLOS ONE

Additional Editor Comments (optional):

Reviewers' comments:

Reviewer's Responses to Questions

**Comments to the Author**

1. If the authors have adequately addressed your comments raised in a previous round of review and you feel that this manuscript is now acceptable for publication, you may indicate that here to bypass the “Comments to the Author” section, enter your conflict of interest statement in the “Confidential to Editor” section, and submit your "Accept" recommendation.

Reviewer #1: (No Response)

2. Is the manuscript technically sound, and do the data support the conclusions?

Reviewer #1: Yes

3. Has the statistical analysis been performed appropriately and rigorously? 

Reviewer #1: Yes

4. Have the authors made all data underlying the findings in their manuscript fully available?

Reviewer #1: Yes

5. Is the manuscript presented in an intelligible fashion and written in standard English?

Reviewer #1: (No Response)

6. Review Comments to the Author

Reviewer #1: (No Response)

7. PLOS authors have the option to publish the peer review history of their article (what does this mean?). If published, this will include your full peer review and any attached files.

Reviewer #1: **Yes: **Saloua Biyada

---

## [Editor Report · Acceptance letter]

22 May 2023

PONE-D-22-30019R2 

Evolution of fungal and non-fungal eukaryotic communities in response to thermophilic co-composting of various nitrogen-rich green feedstocks 

Dear Dr. Matheri:

I'm pleased to inform you that your manuscript has been deemed suitable for publication in PLOS ONE. Congratulations! Your manuscript is now with our production department. 

Kind regards, 

on behalf of

Dr. Sudeshna Bhattacharjya 

Academic Editor

PLOS ONE